# Hi-C calibration by chemically induced chromosomal interactions

Yi Li [ID][1,2✉], Christoph W A Fischer[1,2], Fan Zou[2,3], Manyu Du [ID][1,2] & Lu Bai [ID][1,2,3✉]

## Abstract

The genome-wide chromosome conformation capture method, Hi-C, has greatly advanced our understanding of genome organization. However, its quantitative properties, including sensitivity, bias, and linearity, remain challenging to assess. Measuring these properties in vivo is difficult due to the heterogenous and dynamic nature of chromosomal interactions. Here, using Chemically Induced Chromosomal Interaction (CICI) method, we create stable intra- and inter-chromosomal interactions in G1-phase budding yeast across a broad range of contact frequencies. Hi-C analysis of these engineered cell populations demonstrates that static intra-chromosomal loops do not generate Topologically Associated Domains (TADs) and only promote 3D proximity within 10-60 kb flanking regions. At moderate sequencing depth, Hi-C is sensitive enough to detect interactions occurring in 5-10% of cells. It also shows no inherent bias toward intra- versus inter-chromosomal interactions. Furthermore, we observe a linear relationship between Hi-C signal intensity and contact frequency. These findings illuminate the intrinsic properties of the Hi-C assay and provide a robust framework for its calibration.

Subject Categories Chromatin, Transcription & Genomics; Methods & Resources

## Introduction

Chromosome conformation capture-based methods, such as Hi-C, evaluate spatial proximity between distant chromosomal regions by measuring ligation frequencies (Dixon et al, 2012; Lieberman-Aiden et al, 2009). While Hi-C signals are generally interpreted as "contact frequencies," their quantitative interpretation remains challenging (Lajoie et al, 2015). Specifically, Hi-C-derived contact frequencies cannot be directly converted into the fraction of cells that engage in these interactions, and changes in interaction levels may not result in proportional changes in Hi-C signals. Furthermore, there may be biases in Hi-C signals among loci pairs. For example, it was reported

that inter-chromosomal interactions are underrepresented in Hi-C data compared to intra-chromosomal ones, even though they occur at similar frequencies (Maass et al, 2018).

In principle, Hi-C can be calibrated using imaging techniques such as DNA fluorescence in situ hybridization (FISH), i.e., by visualizing loci pairs and comparing their co-localization probabilities with the corresponding Hi-C signals (Fudenberg and Imakaev, 2017). However, due to limited optical resolution, imaging-based quantification of co-localization usually relies on thresholding-based analyses, making it difficult to precisely determine the fraction of cells where the two loci interact. Live-cell imaging combined with mathematical modeling can overcome some of these limitations (Gabriele et al, 2022), but this approach is inherently low-throughput, and absolute looping probabilities have only been measured for a handful of locus pairs.

To address these issues, we took advantage of chemically induced chromosomal interaction (CICI), a synthetic biology method we previously developed in budding yeast (Du et al, 2022). In this system, LacI-FKBP12 and TetR-FRB fusion proteins are expressed and targeted to integrated LacO and TetO arrays, and the addition of rapamycin induces a stable interaction between the two arrays (Fig. 1A). This setup allows for accurate measurement of the fraction of cells in which the loci pair are in contact, providing a means to calibrate Hi-C signals with absolute contact frequencies.

In this study, we used CICI to generate cell populations with tunable interaction frequencies between selected locus pairs and performed Hi-C on these cells to assess the sensitivity, linearity, and potential biases of the method. Our results show that, with ~18 M contact reads, Hi-C is sensitive enough to detect chromosomal interactions occurring in as few as <10% of cells, and Hi-C signal strength scales linearly with the fraction of interacting cells. Among the loci pairs we examined, Hi-C does not display significant bias toward intra- vs inter-chromosomal interactions. We also find that static intra-chromosomal loops do not lead to the formation of TADs. Finally, by comparing the Hi-C signal at the CICI junction with the genome-wide intensity, we provide an absolute quantification of contact frequencies across the genome. Together, these findings provide insights into the nature of the Hi-C signals and their relation to genomic interactions.

## Results and discussion

The CICI method relies on the expression of four fusion proteins, LacI-FKBP12, TetR-FRB, LacI-GFP, and TetR-mCherry, which

[1]Department of Biochemistry and Molecular Biology, The Pennsylvania State University, University Park, PA 16802, USA. [2]Center for Eukaryotic Gene Regulation, The Pennsylvania State University, University Park, PA 16802, USA. [3]Department of Physics, The Pennsylvania State University, University Park, PA 16802, USA.
✉E-mail: yili452@tsinghua.edu.cn; lub15@psu.edu

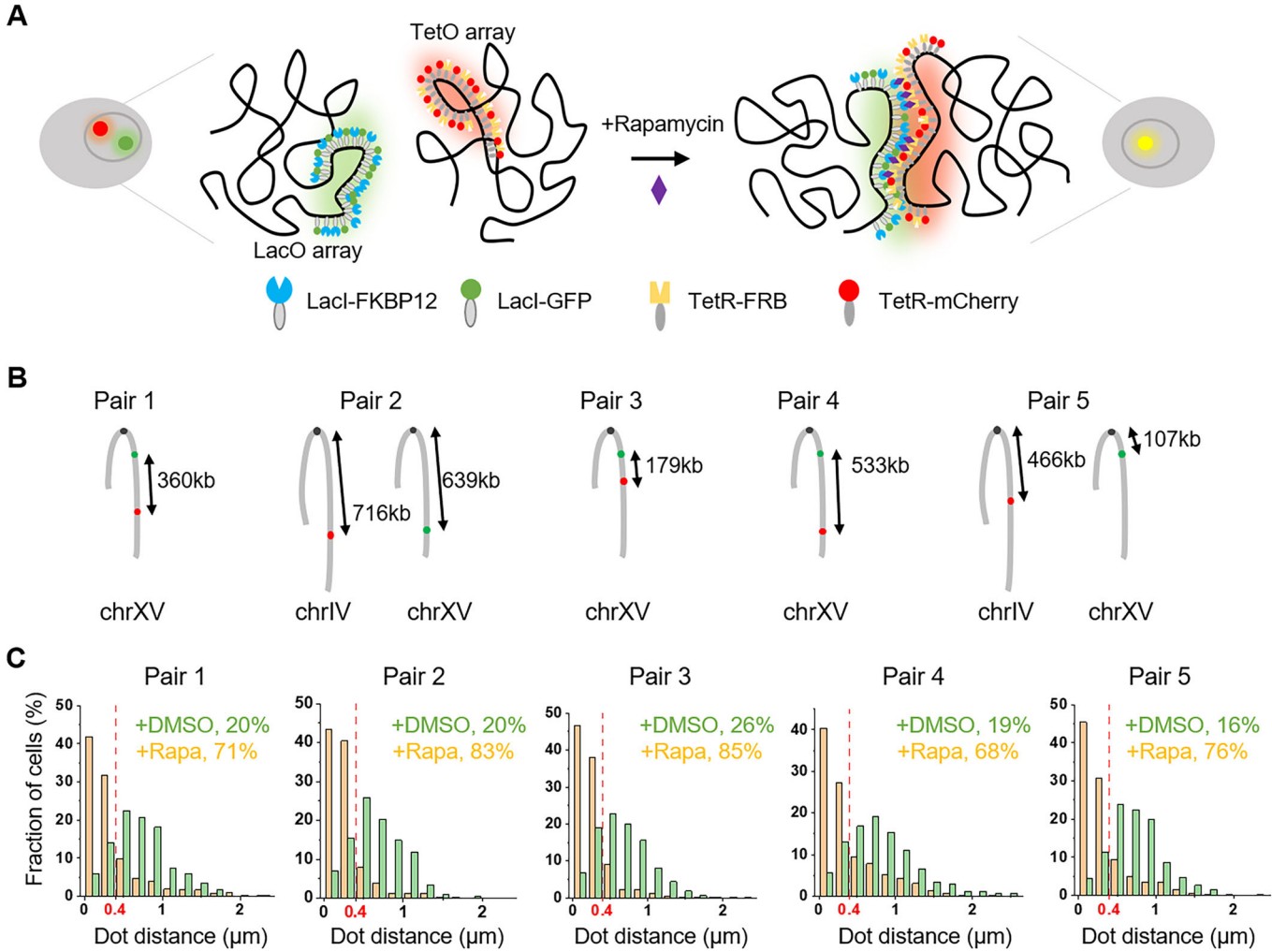

**Figure 1. CICI forces the co-localization of target loci pair.**

(A) Schematic representation of the CICI system. LacO and TetO arrays are inserted into two chromosomal loci, and four fusion proteins are expressed to induce and visualize chromosomal interactions upon rapamycin addition. (B) Configuration of LacO and TetO arrays for CICI pair 1-5. (C) Histograms of dot distances between two loci pairs ± rapamycin. Threshold for co-localization (vertical line): 0.4 μm. Only cells with both dots detectable are included in this analysis. Number of cells analyzed: pair 1 (+rapamycin: $n = 257$, -rapamycin: $n = 286$); pair 2 (+rapamycin: $n = 346$, -rapamycin: $n = 202$); pair 3 (+rapamycin: $n = 1067$, -rapamycin: $n = 822$); pair 4 (+rapamycin: $n = 988$, -rapamycin: $n = 969$); pair 5 (+rapamycin: $n = 1660$, -rapamycin: $n = 571$). Source data are available online for this figure.

bind to inserted LacO and TetO arrays in a rapamycin-resistant yeast strain. Upon rapamycin addition, dimerization of FKBP12 and FRB enables physical association of the two arrays when they come into close proximity (Fig. 1A). LacI-GFP and TetR-mCherry label the two arrays as distinct fluorescent foci, allowing direct visualization of CICI formation (Du et al, 2022). Depending on the genomic insertion sites of the LacO and TetO arrays, CICI can drive either intra-chromosomal interactions (arrays on the same chromosome) or inter-chromosomal interactions (arrays on different chromosomes). In this study, we applied CICI to three intra-chromosomal pairs (1, 3, 4) and two inter-chromosomal pairs (2 and 5) in G1-arrested cells (Fig. 1B). CICI was successfully induced for all loci pairs, as indicated by increased co-localization between the fluorescent foci in the presence of rapamycin (Fig. 1C). Without rapamycin, 16–26% of cells exhibited co-localization with distance between the two arrays below the threshold 0.4 μm. In the

presence of rapamycin, this number increases substantially to 68–85%.

To verify that observed co-localization represents genuine CICI contacts, which are expected to be stable over time (Du et al, 2022), we performed time-lapse imaging and calculated the absolute contact frequency as the fraction of cells that display continuous co-localization over time (Fig. EV1A; Methods). In the absence of CICI induction, co-localization of the two arrays is highly dynamic, and continuous contacts are rare, occurring in only 0.6–4% of cells across the five locus pairs (Fig. 2A). In contrast, rapamycin-induced CICI formation resulted in a much higher frequency of stable contacts, ranging from 62–69% (Fig. 2A). The "net gain" of CICI probability (60–67%) is similar for all pairs, regardless of the LacO/TetO localization, which is consistent with our previous observations (Du et al, 2022). Notably, the time-lapse-based analysis reduces the impact of thresholding on the co-localization estimates.

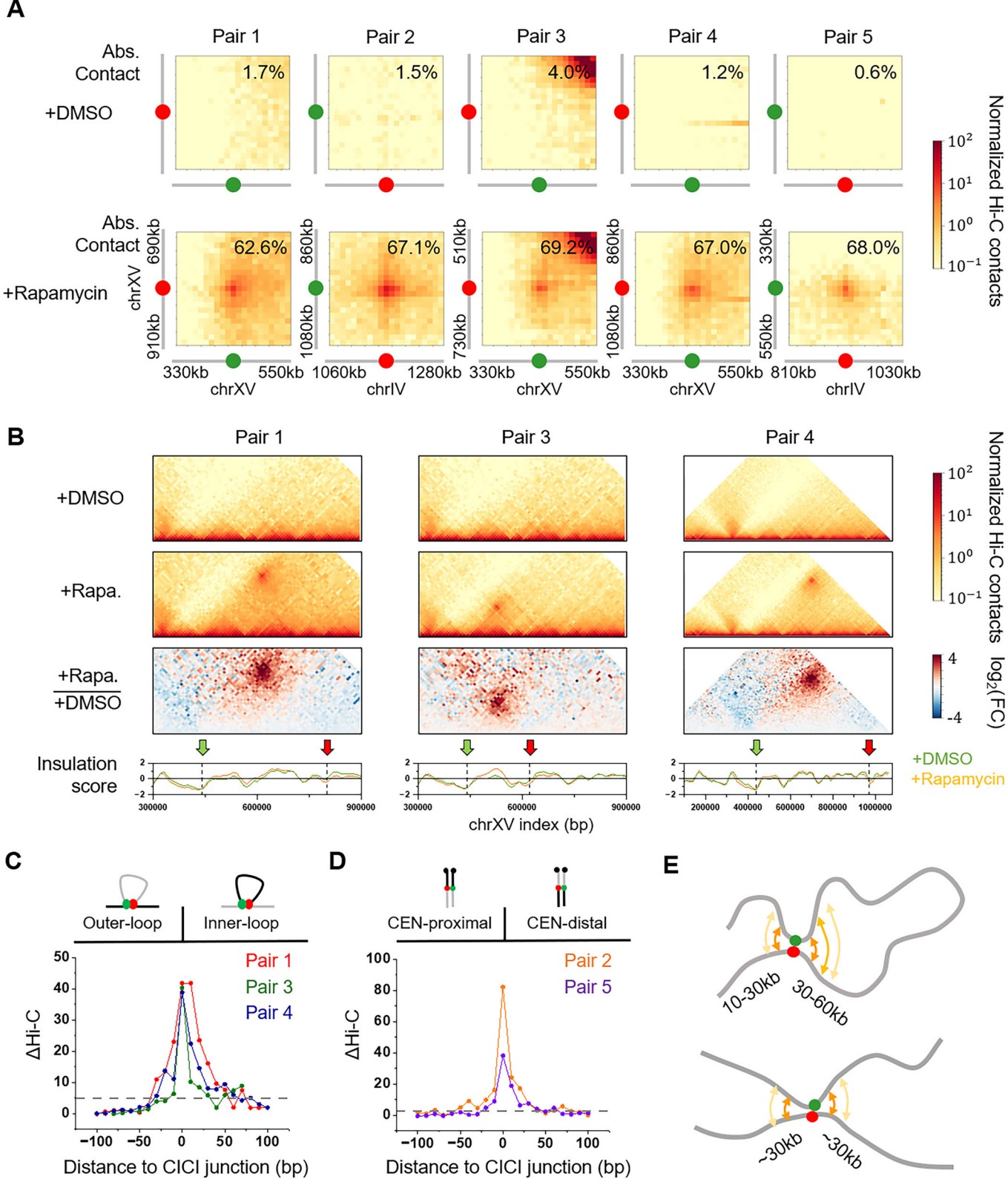

**Figure 2. CICI formation does not induce TAD structure.**

(A) Hi-C contact maps near the CICI junctions (±100 kb) ± rapamycin for all loci pairs. Two to three independent Hi-C replicates are conducted for each pair (same for below). Absolute contact frequencies (fraction of cells exhibiting continuously dot co-localization) are also shown. (B) Hi-C contact maps ± rapamycin and their log2 fold-change near the three intra-chromosomal CICI pairs (red and green arrows). Bottom panel: insulation scores in the corresponding regions. (C, D) Hi-C signal difference ±rapamycin near all CICI pairs. Differences above three standard deviation from the background (marked by the dotted lines) are considered as significant. (E) Cartoon illustrating the range of enhanced proximity from static intra-chromosomal and inter-chromosomal loops. Source data are available online for this figure.

For example, extending the threshold to 0.5 μm only causes mild increases in CICI probabilities for pair 1 from 62.6 to 65.4%.

Next, we performed Hi-C measurements in the same cell populations with and without rapamycin treatment. The formation of CICIs for all loci pairs in the presence of rapamycin is evident from elevated Hi-C signals near the CICI junctions (Fig. 2A). Notably, the increases in local Hi-C contacts for the inter-chromosomal pairs (2 and 5) are comparable to those observed for intra-chromosomal pairs (1, 3, and 4), suggesting that, at least for CICI-mediated interactions, Hi-C does not preferentially detect intra-chromosomal contacts. CICI formation also causes perturbations to the local chromatin architecture. For intra-chromosomal pairs, we observe significantly increased Hi-C signals within a 30–60 kb inner-loop region and a 10–30 kb outer-loop region flanking the junction (Fig. 2B,C). For inter-chromosomal pairs, we observed increased trans-chromosomal interactions spanning ~30 kb regions flanking both sides of the CICI junction (Fig. 2D).

In mammalian cells, chromatin loops are often associated with TADs (Hansen et al, 2018). TAD structures in budding yeast are less well defined (Duan et al, 2010), likely due to the absence of loop-anchoring factors such as CTCF. The strong static loops engineered by CICI could potentially promote the formation of "self-interacting domains" analogous to TADs; however, the short-range effects observed in Fig. 2B,C are far below the length scale of mammalian TADs, which typically span hundreds of kilobases to several megabases (Dixon et al, 2012). To investigate this further, we applied a standard TAD detection algorithm (Ramírez et al, 2018) to calculate the insulation scores for regions surrounding all intra-chromosomal pairs. Although we observed mild dips in insulation near the CICI junctions, these features were indistinguishable with or without rapamycin treatment (Fig. 2B), indicating that they likely arise from reduced Hi-C mappability at the repetitive arrays rather than bona fide domain formation. This finding is consistent with previous polymer simulations showing that TADs emerge from dynamic loop extrusion rather than static loops (Fudenberg et al, 2016). More specifically, during asynchronized loop extrusion, loops form at various locations among individual cells, bringing different regions within TAD boundaries together. The elevated proximity within the entire TADs is thus a consequence of population-averaging, rather than the whole regions forming highly compacted clusters. In contrast, CICI produces bridging for a fixed loci pair, which can only enhance proximity over tens of kilobases near the junction (Fig. 2E). Such engineered CICI contacts may instead resemble natural long-range interactions among polycomb domains or super-enhancers, which are mediated by multivalent protein–protein interactions and are largely independent of CTCF and cohesin (Ogiyama et al, 2018; Rhodes et al, 2020).

To evaluate the sensitivity and linearity of Hi-C signals, we prepared cell populations ± CICI induction for all loci pairs, cross-linked them with formaldehyde, combined them in different ratios, and performed Hi-C measurements (Fig. 3A). This approach allows us

to create artificial cell populations with a wide range of absolute contact frequencies, from ~1 to ~70%. The Hi-C results demonstrate that, with our modest sequencing depth of ~18 M contact reads, interactions in less than 10% of cells can be detected (Figs. 3B and EV1B). We quantified the Hi-C signals with and without bias correction (Imakaev et al, 2012) and plotted them against the corresponding absolute contact frequencies (Fig. 3C and EV1C). Remarkably, Hi-C signals exhibit a largely linear relationship with contact frequencies across the full range. The slopes for the three intra-chromosomal pairs (pair 1, 3, and 4) are similar, whereas one inter-chromosomal pair (pair 2) shows a mildly higher slope and the other (pair 5) a mildly lower slope (Fig. 3C). To better isolate the contribution of the stable, rapamycin-induced CICI interaction, rather than random collisions, to the Hi-C signal, we subtracted the background contact frequency measured in the −rapamycin condition from the +rapamycin signal over the same contact window. Because the Hi-C background is low, this correction has minimal impact on the linearity and slope of Fig. 3C (Fig. EV2). Overall, these data indicate that Hi-C signals are proportional to underlying contact frequencies, with modest variability in sensitivity among loci, and that Hi-C does not intrinsically lack sensitivity for detecting inter-chromosomal interactions.

Using the fitted linear function, we converted the Hi-C signals measured over the native genome into absolute contact frequencies and averaged among regions separated by varying distances (Fig. 3D). Regions within 40 kb show contact frequencies reaching 90%, indicating that contacts in this range can be captured by Hi-C in a large fraction of cells. 40 kb genomic distance corresponds to 250–350 nm in 3D space (Bystricky et al, 2004), agreeing well with the estimated Hi-C capture radius of 100–400 nm (Belaghzal et al, 2017). For regions that are separated by ~700 kb, the interactions on average only occur in ~1% of cells.

Compared with other Hi-C calibration approaches, the synthetic CICI system offers several unique advantages: (1) interactions can be induced between specific locus pairs in a large fraction of cells (>60%), far exceeding the natural looping frequency in the genome (typically ~1%), resulting in a markedly improved signal-to-noise ratio in Hi-C measurements, (2) the stable nature of these interactions allows for reliable quantification of contact frequencies through time-lapse imaging, (3) interactions across different intra- and inter-chromosomal loci pairs can be created through the same mechanism, and (4) by mixing fixed cells with and without CICI at variable ratios, we can precisely modulate contact frequency while maintaining a consistent interaction mechanism. The main limitation of CICI is that it enforces loops via engineered interactions, and the resulting contact configuration and stability may differ from natural genomic interactions and bypass normal regulatory controls. In addition, CICI setup involves tuning the expression of four fusion proteins and integrating two repetitive arrays, which can be labor-intensive.

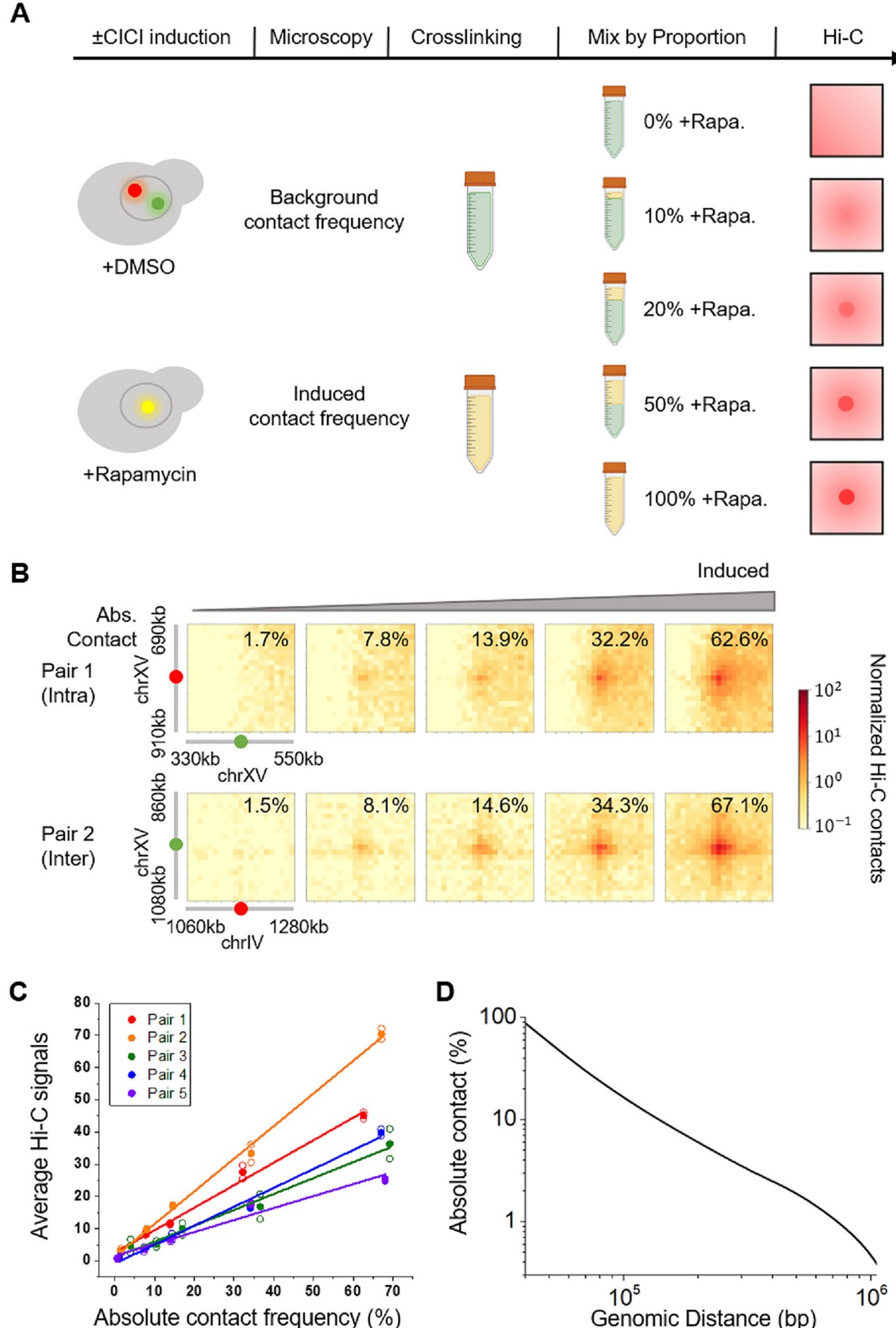

◀ **Figure 3. Hi-C signal scales linearly with CICI contact frequency.**

(A) Experimental workflow. Populations of cells with variable contact frequencies between CICI pairs are generated by mixing fixed cells ± rapamycin at different ratios. (B) Hi-C contact maps for each population in A for pair 1 and 2 (pair 3–5 are shown in Fig. EV1B). ±100 kb regions centered at the CICI junctions are shown. Data were merged from two biological replicates, and the absolute contact frequencies between loci pairs are indicated. Hi-C data at the two end points (0 and 100% + rapamycin populations) are direct repeats from Fig. 2A. (C) Normalized and corrected Hi-C signals as a function of contact frequencies between loci pairs. Hi-C signals are calculated by taking an average of ±10 kb regions centered at the CICI junctions. Hollow and solid dots represent data from individual replicates and their mean, respectively. Lines are linear fits to the data. (D) Hi-C signals as a function of genomic distance. The signals are converted into absolute contact frequencies based on the average slope of the three intra-chromosomal pairs in (C). Source data are available online for this figure.

In this well-controlled system, Hi-C signals display the same linear relationship with contact frequency for both intra- vs inter-chromosomal loci pairs. Therefore, the reported non-linearity between contact frequency and Hi-C signals (Fu et al, 2024), as well as biases towards intra-chromosomal interactions (Maass et al, 2018), are not intrinsic limitations of the Hi-C methodology. Consistent with our observation, a recent study reported a linear relationship between normalized Micro-C signals and the absolute probabilities of natural loops in mouse and human cells (Jusuf et al, 2025). We therefore believe that the linear dependence between Hi-C signal and interaction probability extends to natural chromatin contacts across multiple species.

## Methods

### Reagents and tools table

| Reagent/resource | Reference or source | Identifier or catalog number |
| --- | --- | --- |
| **Experimental models** | | |
| *Saccharomyces cerevisiae* (W303) | From Fred Cross lab: Bai et al, *Dev Cell*, 2010 | |
| **Recombinant DNA** | | |
| pSR11 | Rohner et al, 2008 | |
| pSR13 | Rohner et al, 2008 | |
| pMY63 | Du et al, 2022 | |
| pMY48 | Du et al, 2022 | |
| pSR13-S1 | This study | Table EV1 |
| pSR11-S3 | This study | Table EV1 |
| pSR13-S4 | This study | Table EV1 |
| pSR11-S6 | This study | Table EV1 |
| pSR11-S2 | This study | Table EV1 |
| pSR11-S4 | This study | Table EV1 |
| pSR11-S5 | This study | Table EV1 |
| **Chemicals, enzymes and other reagents** | | |
| Formaldehyde, 37% | Fisher Scientific | RSOF0010-250A |
| Glycine | Fisher Scientific | 194825 |
| Triton X-100 | Sigma-Aldrich | 9036-19-5 |
| Protease Inhibitor Cocktail | Thermo Fisher | 87786 |
| Glass beads, acid-washed | Sigma-Aldrich | G8772 |
| Alpha-factor | Zymo Research | Y1001 |

| Reagent/resource | Reference or source | Identifier or catalog number |
| --- | --- | --- |
| Rapamycin | Sigma-Aldrich | 53123-88-9 |
| DpnII | New England Biolabs | R0543L |
| Klenow | New England Biolabs | M0212 |
| Biotin-14-dATP | Thermo Fisher | 19524016 |
| dNTP set | Thermo Fisher | R0181 |
| T4 DNA ligase | New England Biolabs | M0202L |
| Proteinase K | Thermo Fisher | EO0492 |
| T4 DNA Polymerase | New England Biolabs | M0203 |
| DNA clean and concentrator-5 kit | Zymo Research | D4014 |
| MyOne™ streptavidin C1 beads | Thermo Fisher | 65001 |
| NEBNext Ultra II DNA Library Prep Kit | New England Biolabs | E7645L |
| **Software** | | |
| Hi-C-pro (v3.1.0) | Servant et al, 2015 | |
| HiCExplorer (v3.7.2) | Ramírez et al, 2018 | |
| Cooler (v0.10.0) | Abdennur and Mirny, 2020 | |
| Cooltools (v0.7.0) | Open2C et al, 2024 | |
| **Other** | | |
| Illumina NextSeq 2000 | Illumina | |
| Leica DMI6000 | Leica | |

### Plasmid and strain construction

All the strains used in this study were derived from a W303-based strain, yLB109 (MAT**a**, *tor1-1*, *fpr1::NAT*, *bar1*), for rapamycin resistance. We constructed the CICI strain by integrating (1) plasmids containing LacO or TetO arrays into desired loci pairs, (2) a plasmid containing *LacI-FKBP12* and *TetR-FRB* driven by *REV1pr* into the *ADE2* locus, and (3) a plasmid containing *LacI-GFP* and *TetR-mCherry* driven by *REV1pr* into the *HIS3* locus. All the plasmid and strains used here are listed in Table EV1.

### CICI induction

Yeast strains were grown in 200 mL SCD-Met medium until OD660 reached 0.3. To arrest cells in the G1-phase, 80 µM alpha-factor (Zymo, Y1001) was added to the culture for a total of 2.5 h.

During the final hour of this incubation, 1 ng/μL rapamycin (Sigma, 53123-88-9) was added to induce CICI. After 1 h of rapamycin treatment, cells were harvested for imaging analysis and Hi-C measurement.

## Imaging analysis

We performed time-lapse imaging with a Leica fluorescent microscope (Leica DMI6000 with Hamamatsu ORCA-R2 C10600 camera), We used 7 z-stacks with 0.6 μm spacing for GFP and mCherry channels and 0.12–0.15 s fluorescence excitation exposure was used for each stack. The time spent on each channel is about 18 s. Three consecutive time frames were taken with a time interval of 240 s. We used custom Matlab software developed previously to annotate cells, detect dot positions, and calculate dot distances (Zou and Bai, 2019). Because of the limited axial resolution of our system, we report the inter-dot distances based on their x-y projections. Absolute contact frequency for each loci pair is defined as the fraction of cells that show co-localized dots (x-y distance <0.4 μm) in three consecutive time frames. Such a continuous co-localization criterion is long enough to effectively suppress false positives caused by transient chromosomal encounter (without rapamycin, this co-localization only occurs in 1–2% of cells). It is also short enough to tolerate loss of chromatin dots over time.

## Hi-C experiments

Hi-C was adapted from a previously published protocol (Li et al, 2024). Yeast were incubated in 200 mL SCD-Met medium until OD660 reached 0.3. Cells were fixed with 3% formaldehyde (Fisher, RSOF0010-250A) for 20 min at 25 °C and then quenched with 0.2 M glycine (Fisher, 194825) for 20 min at room temperature. Cells were collected by centrifugation at 3000 rpm for 5 min and washed with SCD-Met. Populations of cells with different absolute contact frequencies between loci pairs were generated by mixing ± rapamycin populations in different proportions. Cell pellets were resuspended in 1 mL TBS (Tris-buffered saline), 1% Triton X-100 (Sigma, 9036-19-5), and 1X protease inhibitor cocktail (Thermo Fisher, 87786). Cell lysis was generated by adding 500 μL acid-washed glass beads (Sigma, G8772) and vortexing for 25 min at 4 °C. The chromatin was recovered through centrifugation at 3000 rpm for 5 min, washed with 1 mL TBS, resuspended in 500 μL 10 mM Tris-HCl buffer and digested with DpnII (NEB, R0543L) overnight at 37 °C. Digested DNA fragments were filled in with biotin-labeled dATP by incubating with Klenow enzyme (NEB, M0212), biotin-14-dATP (Thermo Fisher, 19524016), dCTP, dTTP, and dGTP (Thermo Fisher, R0181) for 4 h at room temperature. The biotin-filled DNA fragments were ligated by T4 DNA ligase (NEB, M0202L) for 4 h at 16 °C. Crosslink was reversed by incubation with proteinase K (Thermo Fisher, EO0492) at 65 °C overnight. DNA was purified by phenol-chloroform extraction. Biotin-labeled, unligated fragment ends were removed by incubating with T4 DNA Polymerase (NEB, M0203), dATP and dGTP for 4 h at 20 °C. DNA was cleaned by the DNA Clean and Concentrator-5 kit (Zymo, D4014) and sheared by Diagenode Bioruptor Pico (EZ mode, 30 s on, 30 s off, 15 cycles). Biotin-labeled DNA was enriched by MyOne™ streptavidin C1 beads (Thermo Fisher, 65001). Hi-C libraries were prepared with the NEBNext Ultra II DNA Library Prep Kit. Next-generation sequencing was performed on NextSeq 2000, and 100 million of 50 bp paired-end reads were generated for each replicate.

## Hi-C data analysis

To perform Hi-C data analysis, Paired-end reads were aligned to the sacCer3 reference genome with Hi-C-pro (version 3.1.0) (Servant et al, 2015). Raw matrix was converted to cool format using hicConvert-Format with HiCExplorer (version 3.7.2) (Ramírez et al, 2018). Samples with different read counts were normalized to the smallest read count using hicNormalize with HiCExplorer (version 3.7.2) (Ramírez et al, 2018), followed by correcting for the number of restriction sites per bin utilizing a fast-balancing algorithm reported earlier (Knight and Ruiz, 2012). Hi-C interaction heatmaps were generated using hicPlotMatrix with HiCExplorer (version 3.7.2) (Ramírez et al, 2018). To plot Hi-C heatmaps, calculate the insulation score, and calculate the difference of Hi-C signals in Fig. 2C, data from different replicates were merged and normalized for read counts. The insulation score in Fig. 2B was calculated using hicFindTADs, with the following parameters: --minDepth 30000 --maxDepth 100000 --step 10000 --thresholdComparisons 0.05 --delta 0.01 --correctForMultipleTesting fdr. To extract Hi-C signals at ±10 kb regions centered at the CICI junctions for different pairs in Fig. 3C, data from individual replicates of different pairs were normalized for read counts and corrected. Hi-C signals of the 30 kb × 30 kb matrix centered at the CICI junction were extracted, and the average was calculated with Cooler (version 0.10.0) (Abdennur and Mirny, 2020) and Cooltools (version 0.7.0) (Open2C et al, 2024). To calibrate the Hi-C signals with the absolute contact frequency in Fig. 3D, data from 0% CICI samples of pairs 1, 3, and 4 (Intra-pairs) were merged to generate the contact vs distance curve.

## Data availability

The sequencing data generated in this study have been deposited into the Gene Expression Omnibus (GEO) database under accession code GSE283767 (https://www.ncbi.nlm.nih.gov/geo/query/acc.cgi?acc=GSE283767).

The source data of this paper are collected in the following database record: biostudies:S-SCDT-10_1038-S44319-026-00772-x.

## Peer review information

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

## Acknowledgements

We thank all members of the Bai lab for insightful comments on the manuscript. We also acknowledge core facilities at the Penn State University, including Genomics (RRID:SCR_023645), Flow Cytometry (RRID:SCR_024460) and Genomics Research Incubator (RRID:SCR_024530) for sequencing and flow cytometry usage. This work is supported by the National Institutes of Health (T32 GM125592 to Y.L. and R35 GM139654 to L.B.) and the National Science Foundation (MCB- 2016266 to L.B.).

## Author contributions

**Yi Li**: Conceptualization; Data curation; Formal analysis; Validation; Investigation; Methodology; Writing—original draft; Project administration; Writing—review and editing. **Christoph W A Fischer**: Data curation; Formal analysis; Investigation; Methodology. **Fan Zou**: Conceptualization; Resources; Software; Methodology. **Manyu Du**: Conceptualization; Supervision; Funding acquisition. **Lu Bai**: Conceptualization; Resources; Supervision; Funding acquisition; Investigation; Writing—original draft; Writing—review and editing.

Source data underlying figure panels in this paper may have individual authorship assigned. Where available, figure panel/source data authorship is listed in the following database record: biostudies:S-SCDT-10_1038-S44319-026-00772-x.

## Disclosure and competing interests statement

The authors declare no competing interests.

# Expanded View Figures

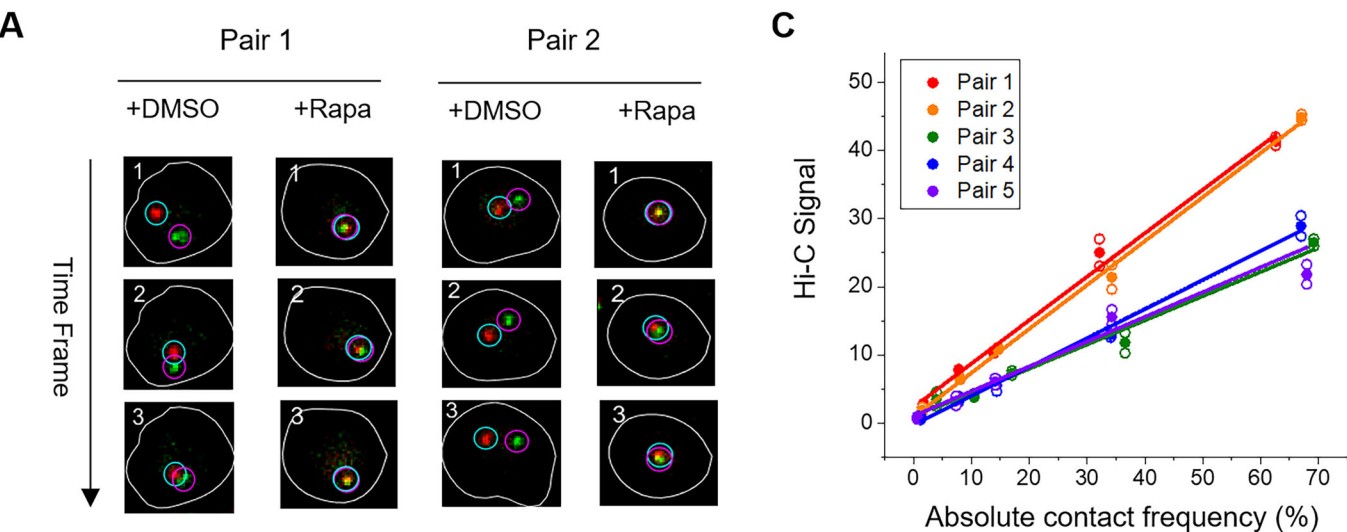

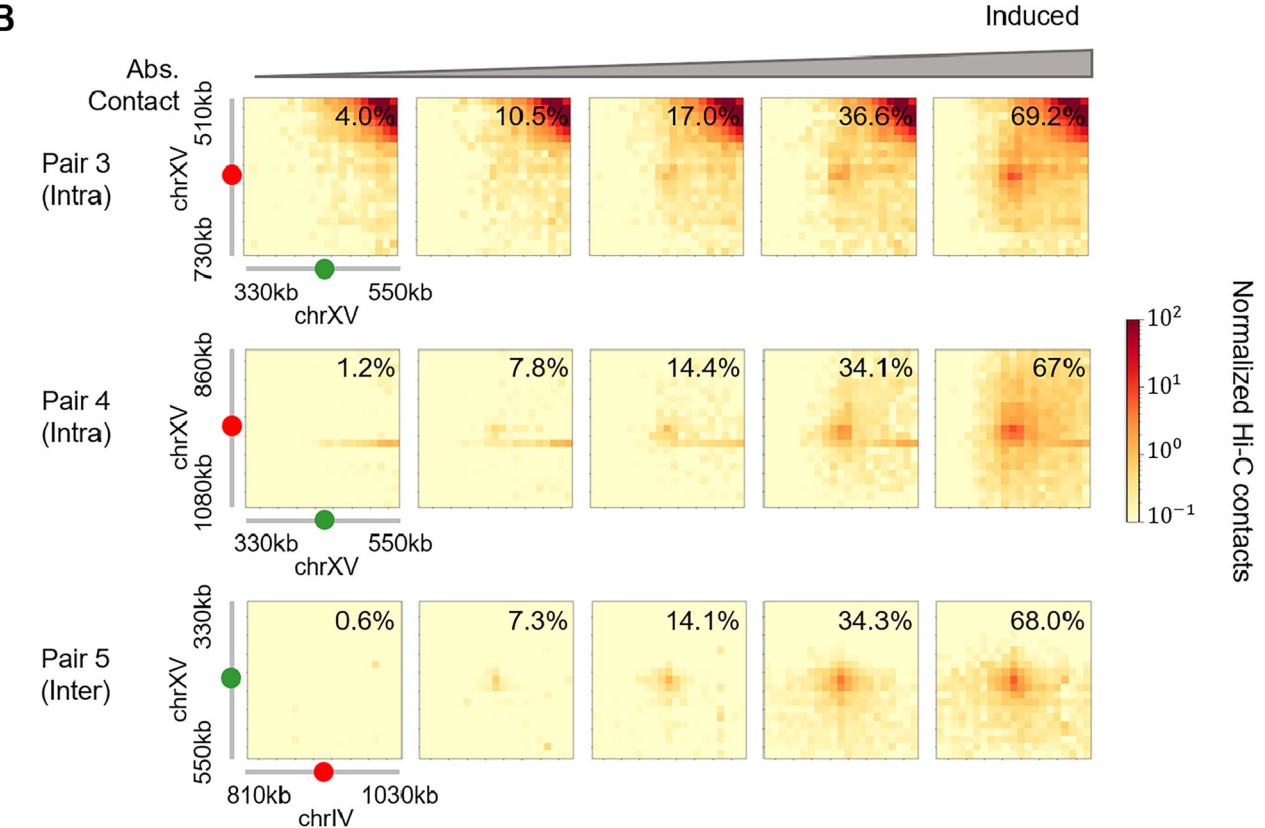

**Figure EV1. Extra imaging and Hi-C data in CICI strains.**

(A) Typical image data for CICI analysis. For each condition, annotated images of three consecutive frames are shown. Loci pairs are considered to be in contact if dots are continuously co-localized. (B) Hi-C data for pair 3–5 with titration of CICI cells. Hi-C data at the two end points (0 and 100% + rapamycin populations) are direct repeats from Fig. 2A. (C) Normalized but not corrected CICI signals as a function of absolute contact frequency.

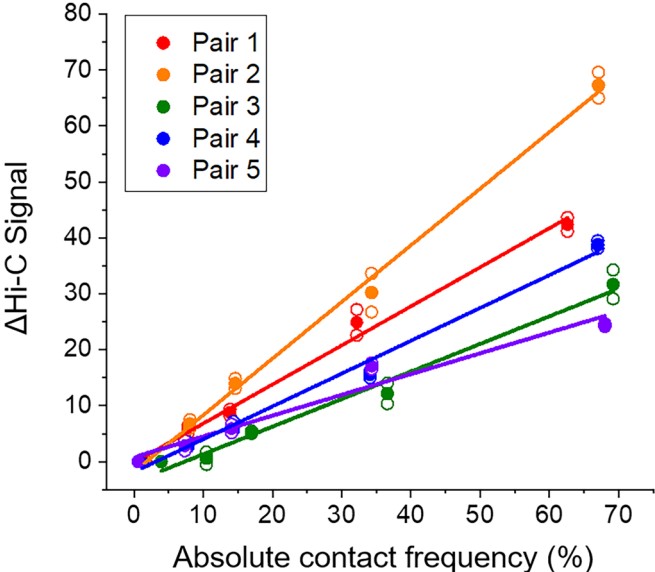

**Figure EV2.  Hi-C signal after background subtraction versus absolute contact frequency.**

This plot is identical to Fig. 3C, except that the −rapamycin Hi-C signals (background) has been subtracted from the +rapamycin Hi-C signals.

