## [Peer Review File · EMBO Reports]

Hi-C Calibration by Chemically Induced Chromosomal Interactions

Yi Li, Christoph Fischer, Fan Zou, Manyu Du, and Lu Bai

Corresponding author(s): Lu Bai (lub15@psu.edu) , Yi Li (yili452@tsinghua.edu.cn)

Review Timeline:

Submission Date:	16th Apr 25
Editorial Decision:	12th Jun 25
Revision Received:	16th Jan 26
Editorial Decision:	16th Mar 26
Revision Received:	19th Mar 26
Accepted:	27th Mar 26

Editor: Esther Schnapp

Transaction Report:

Dear Dr. Bai,

Thank you for the submission of your manuscript to EMBO reports. We have now received the full set of referee reports that is pasted below.

As you will see, the referees acknowledge that the findings are potentially interesting. However, they also have several suggestions for how the data should be strengthened and the study improved. Together, the referees ask for quite extensive revisions, and I would like to know whether you are willing to embark on such a revision process. I think that all referee comments are important and should be addressed, but we can overrule point 3 by referee 3 though this point should be discussed in the ms text. I would like to suggest that you send us a proposed revision plan/point-by-point response that we can discuss, also in a video chat if you like, to see if we can agree on a set of revisions for EMBO reports.

I would thus like to invite you to revise your manuscript with the understanding that the referee concerns must be fully addressed and their suggestions taken on board. Please address all referee concerns in a complete point-by-point response. Acceptance of the manuscript will depend on a positive outcome of a second round of review. It is EMBO reports policy to allow a single round of major revision only and acceptance or rejection of the manuscript will therefore depend on the completeness of your responses included in the next, final version of the manuscript.

We realize that it is difficult to revise to a specific deadline. In the interest of protecting the conceptual advance provided by the work, we recommend a revision within 3 months (12th Sep 2025). Please discuss the revision progress ahead of this time with the editor if you require more time to complete the revisions.

- 1) A data availability section providing access to data deposited in public databases is missing. If you have not deposited any data, please add a sentence to the data availability section that explains that.
- 2) Your manuscript contains statistics and error bars based on $n=2$. Please use scatter blots in these cases. No statistics should be calculated if $n=2$.

5) a complete author checklist, which you can download from our author guidelines . Please insert information in the checklist

that is also reflected in the manuscript. The completed author checklist will also be part of the RPF.

6) Please note that all corresponding authors are required to supply an ORCID ID for their name upon submission of a revised manuscript (). Please find instructions on how to link your ORCID ID to your account in our manuscript tracking system in our Author guidelines

- the name of the statistical test used to generate error bars and P values,
- the number (n) of independent experiments (please specify technical or biological replicates) underlying each data point,
- the nature of the bars and error bars (s.d., s.e.m.),
- If the data are obtained from n {less than or equal to} 2, use scatter blots showing the individual data points.

12) All Materials and Methods need to be described in the main text using our 'Structured Methods' format, which is required for all research articles. According to this format, the Methods section includes a separate Reagents and Tools Table file (listing key reagents, experimental models, software and relevant equipment and including their sources and relevant identifiers) and a Methods and Protocols section describing the methods using a step-by-step protocol format. The aim is to facilitate adoption of the methodologies across labs. More information on how to adhere to this format as well as a downloadable template (.docx) for the Reagents and Tools Table can be found in our author guidelines:

An example of a Method paper with Structured Methods can be found here: <https://www.embopress.org/doi/full/10.1038/s44320-024-00037-6#sec-4>

I look forward to seeing a revised form of your manuscript when it is ready.

Referee #1:

The manuscript by Li et al. presents a novel method for calibrating Hi-C data, using Chemically induced Chromosomal Interaction (CICI) methods to artificially connected chromosomal interactions, further measuring the loci distance, and compared that with Hi-C contacted intensity. Although the manuscript is concise, it is well-structured and presents the data clearly. A particularly compelling aspect of this approach is its foundation in imaging-based calibration, rather than relying on computational or omics-based methods. This imaging-based strategy effectively minimizes potential system biases introduced by sequencing technologies.

Overall, the idea is both innovative and of considerable importance for future calibration of Hi-C and other contact-based omics platforms. I have only two major comments, but they are very important points.

Major comment:

1. Limited sample size.

A major limitation in this study is the small sample size. The authors tested only one pair of inter-chromosomal and one pair of intra-chromosomal interactions. This is insufficient to generalize the applicability and robustness of their calibration system. I strongly recommend that the authors include additional inter-chromosomal and three intra-chromosomal interaction pairs, including pairs separated by less than 40 Kb (see third comment below) to strengthen their conclusions.

2. The claimed linear relationship between Hi-C signal intensity and contact frequency across the genome (e.g., Fig. 2D) is derived from two pseudo-populations. It is good to see that both tested pairs fall on the same linear line, but it would be good to validate the proposed linearized standard curve further: the authors should apply it to additional real interaction datasets to examine how well Hi-C signal intensity correlates with the percentage of cells exhibiting a contact.

3. In Figure 2D the authors scale Hi-C data vs. absolute contact frequencies. This is very interesting but raises a question: Why do loci separated by 40 kb or less all display a 100% contact frequency. That is not consistent with the fact that in Hi-C loci separated by 10 or 20 Kb interact much more frequently than loci separated by 40 Kb. That means that for pairs of loci separated by less than 40 kb there is no longer a relationship between Hi-C and calculated absolute contact frequency (they are all in contact in 100% of the cells, but in Hi-C they differ a lot in interaction frequency). The authors should explore that deeper, as it may indicate that the CICI method overestimates contact frequency.

Minor comment:

1. The pros and cons of different calibrations towards to Hi-C or contact detection systems can be compared in the discussion part together with CICI.

Referee #2:

HiC is an omics method for studying the entire genome organization at the cell population level. In particular, this approach allows detection of large-scale organization levels such as compartments (A/B), but also more localized structuration as chromatin loops mediated by the cohesin complex. However, the quantitative interpretation of these signals, and more specifically the correspondence between an interaction frequency detected by Hi-C and its absolute abundance within the

population, remained unclear.

This article proposes to clarify this link using a controlled system called CICI (Chemically Induced Chromosomal Interaction) for the induction of stable intrachromosomal or interchromosomal contacts *in vivo* and calibrate the Hi-C method. More precisely, two loci proximity relies on the rapamycin dependent interaction between the chimeric proteins LacI-FKBP12 and TetR-FRB respectively associated to LacO or TetO arrays introduced at different positions in the genome. Expression of the LacI-GFP and TetR-mCherry proteins allows to simultaneously monitor the localization of these two loci in the nucleus by microscopy, and to estimate the proportion of cells in which the two sites are stably associated. With this system, the authors prepared cellular populations in which the LacO-TetO arrays are associated in proximity in known and various proportions, to assess the ability of Hi-C method to detect and restore a proportional contact signal. This study nicely shows that for a stable DNA-DNA proximity, the Hi-C signal is linear with respect to the absolute frequency of this contact within the population. However, a few points must be addressed to improve the manuscript.

1) In the table of strains, the *frp1* mutation is not mentioned in the genotypes. In addition, in the sentence "All the strains used in this study were derived from a background strain carrying *tor1* and *fpr* mutations for rapamycin resistance", I guess *fpr* meant *fpr1*? In addition, which yeast background is used?

2) I missed why cells are grown without methionine. The reason should be developed.

3) For clarification, in the sentence 'We used seven z-stacks with 0.6 μm spacing for the GFP and mCherry channels, with fluorescence excitation of 0.12-0.15 s', does 0.6 μm refer to the distance between two z-positions? Does 0.6 μm refer to the distance between two Z positions? If so, this is a large step; most papers addressing foci colocalisations in the nucleus of *S. cerevisiae* use an axial (z) step of 200 nm. Additionally, 'The time spent on each channel is about 18 s.' From this sentence, I understand that images were acquired by performing one z-stack for each channel.

To address colocalisation in living cells, it is better to acquire both channels at each z-position, particularly given that acquiring a channel takes 18 seconds. I would also advise the authors to redo the estimation of the absolute contact frequency using a smaller step size (around 200 nm) to obtain a more accurate estimation of stable pair association. This is important for Hi-C calibration and the establishment of the correspondence between 'absolute contact frequency' and the genomic distance (Fig 2C and 2D).

4) This study is based on characterising the Hi-C methodology using microscopy. Therefore, the analysis of Hi-C data should be more detailed. The reference genome used for read alignment must be specified. The bin size for contact map visualisation in Figures 1D, 1G and 2B should also be specified in the legend. For panel 2C, how are the averaged Hi-C contacts estimated? Is it a sum of the contacts for the 300 kb square windows in Fig. 2B? Are all contact maps subsampled to the same number of contacts for fair comparison?

5) The authors should also explain the rationale behind the methodology used to detect the fraction of cells displaying continuous colocalisation over time. Is the "Three consecutive time frames were taken with a time interval of 240 s" chosen arbitrarily?

6) In the methods, the centrifugation step after cell lysis is not detailed, strength and time should be specified.

7) To strengthen the correspondence between absolute contact and genomic distance, it would be convincing to estimate the real 'absolute contact frequency' by microscopy between a middle-range distance LacO-separated TetO region, and see if the correspondence is correct. I suggest the authors to perform the experiment.

8) I analysed some of the FASTQ files corresponding to *pair1_intra_0*, *pair1_intra_100*, *pair2_inter_0_pairs* and *pair2_inter_100_pairs* from the first replicate. I observed strong differences in genome organisation between pair 1 and 2. For pair 2 in particular, the maps and distance law show greater compaction at medium range and more intrachromosomal contacts than for pair 1. It is as if the pair 2 culture was not properly synchronized in G1, possibly polluted by G2 cells that exhibit such compaction. The authors should present the FACS profiles of the cell cultures used to confirm that the cell synchronisations are correct. Alternatively, images acquired by microscopy could also be helpful because the shape of the cells is a partial indicator of the stage of the cell cycle. This is important because compaction alters both the contact frequency detected at the 300 kb distance range separating LacO-TetO sequences and interchromosomal contacts.

Referee #3:

This study presents a novel calibration of the Hi-C assay using the Chemically Induced Chromosomal Interaction (CICI) method in budding yeast, enabling precise control and quantification of chromosomal contact frequencies. The authors engineered yeast strains with LacO and TetO arrays at specific genomic loci and expressed fusion proteins (LacI-FKBP12, TetR-FRB, LacI-GFP, and TetR-mCherry) to induce and visualize chromosomal interactions upon rapamycin treatment, allowing accurate measurement of contact frequencies via time-lapse microscopy. They demonstrate that Hi-C can sensitively detect interactions

occurring in as few as 7.8% of cells and exhibits a linear relationship between signal intensity and contact frequency across a wide range. Contrary to prior assumptions, Hi-C shows no inherent bias between intra- and inter-chromosomal interactions and static loops do not induce Topologically Associated Domains (TADs). These findings clarify the quantitative capabilities of Hi-C and suggest that previously reported biases likely stem from complexities in native chromatin dynamics rather than limitations of the assay itself.

The technical setup is thoughtfully designed, with clear rationale and implementation of the CICI system. The use of time-lapse microscopy to quantify absolute contact frequencies is a strong complement to the Hi-C data, and the mixing experiments to simulate varying contact frequencies are particularly elegant. The authors also provide sufficient methodological detail to enable reproducibility.

Overall, the manuscript is interesting and presents valuable information and resources for the field. The use of a synthetic, controllable system to benchmark Hi-C performance is innovative and provides a much-needed framework for interpreting Hi-C data quantitatively. The methodology is clearly described, and the results are well-supported by both imaging and sequencing data. This work will be of broad interest to researchers studying genome architecture and chromatin dynamics.

Major comments:

1. Clarification on TAD Formation in Yeast

The authors should improve the interpretation of their data regarding loop formation and TADs. TADs, as classically defined in mammalian systems, are not typically observed in budding yeast, where chromatin is organized into much smaller self-interacting domains (PMID: 26119342). Given this, it is unclear why the authors expected that forcing a long-range loop in yeast would result in TAD formation. This expectation should be better justified or revised in the discussion.

2. Nature of the Synthetic Constructs

The TetO and LacO arrays used in this study are relatively large DNA elements (~3.6 kb and ~6.9 kb, respectively), which differ significantly from the short DNA motifs (e.g., CTCF binding sites) that mediate loop and TAD formation in mammalian genomes. These large synthetic arrays may not recapitulate the fine-scale architectural features of endogenous loop anchors. The authors should discuss how the size and nature of these constructs might influence the observed chromatin interactions and why the resulting contacts resemble long-range interactions seen in polycomb-mediated domains (PMID: 30008320; PMID: 31968256) or metalloops (PMID: 37536338).

3. Extension to Mammalian Systems

While the yeast system is well-controlled and informative, the authors might consider extending this approach to mammalian or other metazoan cells where TADs are more clearly defined. Although this may be beyond the scope of the current study, such experiments would provide valuable insights into whether the synthetic interactions observed here behave more like polycomb-mediated contacts or canonical chromatin domains in higher eukaryotes.

Minor Comments

4. Terminology Clarification: The manuscript occasionally uses the term "TAD" in the context of budding yeast, where such domains are not well-established. It would be helpful to clarify this distinction early in the text and consistently refer to "self-interacting domains" or "domain-like structures" when discussing yeast chromatin organization.

5. Discussion of Limitations: While the authors briefly mention the advantages of the CICI system, a more explicit discussion of its limitations—such as the artificial nature of the constructs and potential off-target effects of rapamycin—would strengthen the manuscript.

We thank all three reviewers for carefully reading our manuscript and providing thoughtful feedback. We are encouraged by the positive comments and have revised the manuscript accordingly. Below is a detailed point-by-point response, with **reviewer comments shown in black** and **our replies in blue**. In our responses, all sentences describing changes made to the manuscript are underlined. **In the revised manuscript, the corresponding changes are highlighted in red.**

Referee #1:

The manuscript by Li et al. presents a novel method for calibrating Hi-C data, using Chemically induced Chromosomal Interaction (CICI) methods to artificially connected chromosomal interactions, further measuring the loci distance, and compared that with Hi-C contacted intensity. Although the manuscript is concise, it is well-structured and presents the data clearly. A particularly compelling aspect of this approach is its foundation in imaging-based calibration, rather than relying on computational or omics-based methods. This imaging-based strategy effectively minimizes potential system biases introduced by sequencing technologies.

Overall, the idea is both innovative and of considerable importance for future calibration of Hi-C and other contact-based omics platforms. I have only two major comments, but they are very important points.

We thank the reviewer for the positive assessment of the paper.

Major comment:

1. Limited sample size.

A major limitation in this study is the small sample size. The authors tested only one pair of inter-chromosomal and one pair of intra-chromosomal interactions. This is insufficient to generalize the applicability and robustness of their calibration system. I strongly recommend that the authors include additional inter-chromosomal and three intra-chromosomal interaction pairs, including pairs separated by less than 40 Kb (see third comment below) to strengthen their conclusions.

We thank the reviewer for this helpful suggestion, as the additional measurements led us to refine our previous conclusions. We performed Hi-C experiments using three additional CICI pairs (Pairs 3–5; see Panel A). Pairs 3 and 4 are intra-chromosomal, with linear genomic separations of 179 kb and 533 kb, respectively, whereas Pair 5 is inter-chromosomal. We agree with the reviewer that testing an intra-chromosomal pair separated by ~40 kb would be of interest. However, such a locus pair is expected to have an average spatial separation of ~200 nm (Bystricky *et al*, 2004), which is well below our co-localization resolution limit of 400 nm. Pair 3 approaches the shortest genomic distance at which CICI efficiency can be reliably quantified by imaging.

Because Yi has since left the lab and a new student is collecting data, we first carried out sanity check to make sure the previous data can be reproduced. We repeated the Hi-C measurements for Pairs 1 under the 0% and 100% CICI condition, as well as Pair 2 under the 100% CICI condition. The new measurements are in good agreement with the original data (Panel B), confirming the reproducibility of these experiments. We also performed imaging-based co-localization analysis for

Pairs 3–5 in the presence or absence of rapamycin. Comparison of the three intra-chromosomal pairs (pair 1, 3, and 4) in the absence of rapamycin (0% CICI) shows that the co-localization probability decreases with increasing linear genomic distance (Panel C), consistent with our expectations.

We then carried out full Hi-C measurements for Pairs 3–5 while titrating the CICI population at 0, 10, 20, 50, and 100%. To ensure consistency, raw Hi-C data for all pairs were processed with the same analysis pipeline with bin size of 10 kb, and the CICI Hi-C signal is calculated as the total contacts within 30 kb centered around the CICI junction. The Hi-C signal vs absolute contact frequency are shown in Panel D. In all cases, Hi-C signals scale linearly with the absolute contact frequencies. However, the Pair3-5 exhibit slopes ~two-fold lower than those observed for Pairs 1 & 2. We reasoned that these differences may arise from locus-dependent variation in DpnII cutting frequency near CICI junctions, leading to technical biases where genomic bins differ in their effective “visibility” in Hi-C, which was well-documented in literature (Imakaev *et al*, 2012). To account for this, we

corrected the data using a previously published matrix-balancing algorithm (Knight & Ruiz, 2012) (Panel E). Following correction, the three intra-chromosomal pairs (1, 3, and 5) show similar slopes, while Pair 2 has a higher slope and Pair 5 has a slightly lower one.

We do not fully understand this slope differences; one possibility is that the assumption in the correction that all genomic bins should have the same total interactions may not be strictly true. Nevertheless, these data confirm our previous conclusion that 1) Hi-C signal intensity scales linearly with true contact frequency, and (2) intra-chromosomal Hi-C signals are not necessarily stronger than inter-chromosomal signals.

In light of these new data, we have substantially reorganized the figures. The revised manuscript now contains three figures instead of two, incorporating additional analyses for the new pairs. The text has been updated accordingly. In addition, the titrated Hi-C heatmap for pair 3–5 has been added as Supplementary Figure 1.

2. The claimed linear relationship between Hi-C signal intensity and contact frequency across the genome (e.g., Fig. 2D) is derived from two pseudo-populations. It is good to see that both tested pairs fall on the same linear line, but it would be good to validate the proposed linearized standard curve further: the authors should apply it to additional real interaction datasets to examine how well Hi-C signal intensity correlates with the percentage of cells exhibiting a contact.

Yeast lacks “loop anchor” proteins like CTCF and therefore does not display highly stable chromatin contacts (Hi-C maps of the WT yeast show few, if any, distinct dot-like features). However, a recent study from the Hansen lab compared Micro-C signals with the absolute probabilities of natural loops, primarily between CTCF sites, in mouse and human cells (Jusuf *et al*, 2025). They reported “a strong linear relationship” between normalized Micro-C signal and looping probability, a result that is fully consistent with our own observations. We therefore believe that the linear dependence between Hi-C signal and interaction probability extends to natural chromatin contacts across multiple species.

We made a comment on this on page 6-7.

3. In Figure 2D the authors scale Hi-C data vs. absolute contact frequencies. This is very interesting but raises a question: Why do loci separated by 40 kb or less all display a 100% contact frequency. That is not consistent with the fact that in HI-C loci separated by 10 or 20 Kb interact much more frequently than loci separated by 40 Kb. That means that for pairs of loci separated by less than 40 kb there is no longer a relationship between Hi-C and calculated absolute contact frequency (they are all in contact in 100% of the cells, but in Hi-C they differ a lot in interaction frequency). The authors should explore that deeper, as it may indicate that the CICI method overestimates contact frequency.

We thank the reviewer for this comment. Based on the new data and analysis, we have replotted Figure 3D using the updated slope derived from Figure 3C. Because this plot focuses exclusively on intra-chromosomal interactions, we calibrated the curve using the average slope of the three intra-chromosomal CICI pairs, which are highly consistent with one another. The resulting curve

(shown below) is similar to the original version, with the Hi-C contact reaches 90% at 40 kb.

To understand this value, note that our Hi-C analysis uses bin size of 10kb and averaging contact frequency within a 30 kb X 30 kb block when calculating the Hi-C signal at the CICI junction (we also tested a finer bin size of 1 kb. However, after correction, the data became excessively noisy, and analysis of the uncorrected 1-kb data yielded results similar to those obtained with our original approach). With the 10kb resolution, a literal interpretation of these data is that two distal 30-kb

genomic blocks, when forced to interact with ~90% efficiency at their centers, will generate a similar number of Hi-C contacts as two 30-kb blocks that are *in cis* and separated by 40 kb. The nature of “contacts” in these two cases is likely to be distinct: the former reflects protein-mediated genomic interactions, whereas the latter likely arises from physical proximity imposed by polymer connectivity. Nonetheless, the effective ligation efficiencies in Hi-C appear to be comparable in these two scenarios.

Minor comment:

1. The pros and cons of different calibrations towards to Hi-C or contact detection systems can be compared in the discussion part together with CICI.

Following this advice, we rewrote the one paragraph in the discussion on page 6: Compared with other Hi-C calibration approaches, the synthetic CICI system offers several unique advantages: 1) interactions can be induced between specific locus pairs in a large fraction of cells (>60%), far exceeding the natural looping frequency in the genome (typically ~1%), resulting in a markedly improved signal-to-noise ratio in Hi-C measurements, 2) the stable nature of these interactions allows for reliable quantification of contact frequencies through time-lapse imaging, 3) interactions across different intra- and inter-chromosomal loci pairs can be created through the same mechanism, and 4) by mixing fixed cells with and without CICI at variable ratios, we can precisely modulate contact frequency while maintaining a consistent interaction mechanism. The main drawback of the CICI system is that it involves elaborate genetic steps to set up, as tuning the expression of four fusion proteins and integrating two repetitive arrays can be labor-intensive.

Referee #2:

HiC is an omics method for studying the entire genome organization at the cell population level. In particular, this approach allows detection of large-scale organization levels such as compartments (A/B), but also more localized structuration as chromatin loops mediated by the cohesin complex. However, the quantitative interpretation of these signals, and more specifically the correspondence

between an interaction frequency detected by Hi-C and its absolute abundance within the population, remained unclear.

This article proposes to clarify this link using a controlled system called CICI (Chemically Induced Chromosomal Interaction) for the induction of stable intrachromosomal or interchromosomal contacts in vivo and calibrate the Hi-C method. More precisely, two loci proximity relies on the rapamycin dependent interaction between the chimeric proteins LacI-FKBP12 and TetR-FRB respectively associated to LacO or TetO arrays introduced at different positions in the genome. Expression of the LacI-GFP and TetR-mCherry proteins allows to simultaneously monitor the localization of these two loci in the nucleus by microscopy, and to estimate the proportion of cells in which the two sites are stably associated. With this system, the authors prepared cellular populations in which the LacO-TetO arrays are associated in proximity in known and various proportions, to assess the ability of Hi-C method to detect and restore a proportional contact signal. This study nicely shows that for a stable DNA-DNA proximity, the Hi-C signal is linear with respect to the absolute frequency of this contact within the population. However, a few points must be addressed to improve the manuscript.

We thank the reviewer for the positive assessment of the paper.

1) In the table of strains, the *frp1* mutation is not mentioned in the genotypes. In addition, in the sentence "All the strains used in this study were derived from a background strain carrying *tor1* and *fpr* mutations for rapamycin resistance", I guess *fpr* meant *fpr1*? In addition, which yeast background is used?

The common strain background is W303-based strain, yLB109 (*MAT_a*, *tor1-1*, *fpr1::NAT*, *bar1*). This information is now included in the manuscript on page 7. As we carried out measurements in more strains, we also added a Table listing all the plasmid and strain information.

2) I missed why cells are grown without methionine. The reason should be developed.

For imaging studies, we typically use synthetic complete media (SCD) rather than rich media (YPD) to reduce background autofluorescence. In our lab, SCD is routinely prepared either without methionine (SCD–Met) or supplemented with saturating methionine (SCD + 10x Met), as many of our lab strains carry constructs under the control of the *MET3* promoter, allowing gene expression to be regulated by methionine availability. Importantly, methionine is not an essential amino acid for budding yeast, and ± Met does not significantly impact the growth rate of wild-type strains. For the CICI strains in this study, methionine is not expected to have any effects, and we used SCD–Met primarily for consistency and convenience across experiments.

3) For clarification, in the sentence 'We used seven z-stacks with 0.6 μm spacing for the GFP and mCherry channels, with fluorescence excitation of 0.12-0.15 s', does 0.6 μm refer to the distance between two z-positions? If so, this is a large step; most papers addressing foci colocalisations in the nucleus of *S. cerevisiae* use an axial (z) step of 200 nm. Additionally, 'The time spent on each

channel is about 18 s.' From this sentence, I understand that images were acquired by performing one z-stack for each channel.

To address colocalisation in living cells, it is better to acquire both channels at each z-position, particularly given that acquiring a channel takes 18 seconds. I would also advise the authors to redo the estimation of the absolute contact frequency using a smaller step size (around 200 nm) to obtain a more accurate estimation of stable pair association. This is important for Hi-C calibration and the establishment of the correspondence between 'absolute contact frequency' and the genomic distance (Fig 2C and 2D).

We thank the reviewer for these thoughtful suggestions. Regarding the z-step size, we currently use an epifluorescence microscope, which has a relatively large depth of field (~1.6 μm) compared to confocal systems (~0.5 μm). Given the typical diameter of a haploid yeast cell (~4–5 μm), a step size of 0.6 μm allows us to traverse the whole yeast within 7 steps, while catching the chromatin dot signals in 2-3 z slices. Because of the limited axial resolution of our system, we report the inter-dot distances based on their projection in the x–y plane. We recognize that this approach introduces some false-positive co-localizations, particularly for dots that are aligned in x–y but separated in z. However, we mitigate this limitation by defining co-localization based on persistence across three consecutive time points, which reduces the chance of misidentifying transient overlaps. We made this clearer on page 8 in the Methods section.

As the reviewer correctly noted, we currently acquire all z-stacks for the GFP channel before proceeding to the mCherry channel. While we agree that alternating acquisition at each z-plane would reduce the temporal offset and improve co-localization accuracy, this is technically challenging with our current setup. Channel switching in our microscope requires mechanical rotation of the filter wheel, which is relatively slow and prone to synchronization issues during time-lapse image acquisition. In our experience, imaging complete stacks sequentially for each channel is much faster and more reliable. We are aware of alternative solutions, such as LED light source or dual-camera setup, that would improve temporal resolution, but these are not currently available to us. We appreciate the reviewer's recommendation and will consider it in future work as we upgrade our imaging capabilities.

4) This study is based on characterising the Hi-C methodology using microscopy. Therefore, the analysis of Hi-C data should be more detailed. The reference genome used for read alignment must be specified. The bin size for contact map visualisation in Figures 1D, 1G and 2B should also be specified in the legend. For panel 2C, how are the averaged Hi-C contacts estimated? Is it a sum of the contacts for the 300 kb square windows in Fig. 2B? Are all contact maps subsampled to the same number of contacts for fair comparison?

We provided a more detailed Hi-C data analysis explanation on page 9 in the Method section.

5) The authors should also explain the rationale behind the methodology used to detect the fraction of cells displaying continuous colocalisation over time. Is the "Three consecutive time frames were

taken with a time interval of 240 s" chosen arbitrarily?

Two freely diffusing chromosome loci can transiently encounter by chance. To distinguish real CICI formation from random encounters, we need to score continuous co-localization across multiple time points. In -rapamycin controls, continuous co-localization across three consecutive frames occurs in only ~1–2% of cells, indicating that this threshold effectively suppresses false positives.

In theory, we can increase stringency by requiring longer co-localization. However, chromatin dots tend to disappear over time after CICI formation. In our design, LacI–GFP and TetR–mCherry are co-expressed for visualization, and they will compete with LacI–FKBP12 and TetR–FRB for binding. Before CICI formation, LacI–GFP and LacI–FKBP12 should have similar binding affinities, so each should occupy ~half of the LacO sites (similar for TetR–mCherry and TetR–FRB on TetO). After CICI formation, however, FKBP12–FRB dimerization would give these fusion proteins a proximity-driven avidity advantage: when LacI–FKBP12 (or TetR–FRB) transiently dissociates from the binding site, it remains locally retained via its partner and therefore can rebind rapidly. This does not happen for LacI–GFP and TetR–mCherry, so they are competed off the arrays over time. Consistently, we observe progressive loss of chromatin dots + rapamycin in CICI strains (but not in -rapamycin control). The requirement of longer co-localization would therefore lose true CICI events and bias the statistics toward underestimation. We add some of these explanations to the Methods section on page 8.

6) In the methods, the centrifugation step after cell lysis is not detailed, strength and time should be specified.

This information is now included in the manuscript on page 8.

7) To strengthen the correspondence between absolute contact and genomic distance, it would be convincing to estimate the real 'absolute contact frequency' by microscopy between a middle-range distance LacO-separated TetO region, and see if the correspondence is correct. I suggest the authors to perform the experiment.

We thank the reviewer for this important point. This comment overlaps with Reviewer #1, Question #1. We now performed the CICI and the corresponding Hi-C measurements on three more LacO-TetO pairs. Please refer to our response there for our new data.

8) I analysed some of the FASTQ files corresponding to pair1_intra_0, pair1_intra_100, pair2_inter_0_pairs and pair2_inter_100_pairs from the first replicate. I observed strong differences in genome organisation between pair 1 and 2. For pair 2 in particular, the maps and distance law show greater compaction at medium range and more intrachromosomic contacts than for pair 1. It is as if the pair 2 culture was not properly synchronized in G1, possibly polluted by G2 cells that exhibit such compaction. The authors should present the FACS profiles of the cell cultures used to confirm that the cell synchronisations are correct. Alternatively, images acquired by microscopy could also be helpful because the shape of the cells is a partial indicator of the stage of the cell cycle. This is

important because compaction alters both the contact frequency detected at the 300 kb distance range separating LacO-TetO sequences and interchromosomal contacts.

We thank the reviewer for the careful and thoughtful evaluation of our manuscript. Following the reviewer's suggestion, we repeated the α -factor G1 arrest in a CICI strain (pair 1) for 2.5 hrs and performed FACS analysis to assess cell cycle distribution. The G1 population increased substantially from ~33% to ~80%, as shown below, although approximately 20% of cells remained in G2/M. This is not too surprising given the doubling time of the strain is between 2.5-3 hrs. We chose not to extend the arrest period further because prolonged α -factor treatment can lead to partial recovery and resumption of the cell cycle in some cells. In addition, extended arrest tends to produce abnormally large cells, which can be less favorable for CICI formation and detection. These results suggest that some variation in the G1 fraction may exist between experiments, which could contribute to the observed differences in Hi-C background, particularly in overall contact compaction.

We also examined the contact-versus-distance relationship across all of our Hi-C datasets. Consistent with the reviewer's observation, we found that pair 2 exhibits higher contact frequencies at distances < 50 kb. Interestingly, all newly collected datasets by Christoph (our new student), including pairs 3-5, pair1_3, and pair2_3, fall between the original pair 1 and pair 2 curves. In particular, pair1_3 and 2_3 are nearly indistinguishable (lower left panel), indicating that there are no fundamental differences among these strains and the previously observed differences are more likely due to cell-cycle arrest conditions. Importantly, the Hi-C signals at the CICI junction are highly similar between the old and new

measurements for pair 1 and pair 2 (panel B on page 2 of the rebuttal). We therefore conclude that these variations do not affect the main conclusions of our study.

Referee #3:

This study presents a novel calibration of the Hi-C assay using the Chemically Induced Chromosomal Interaction (CICI) method in budding yeast, enabling precise control and quantification of chromosomal contact frequencies. The authors engineered yeast strains with LacO and TetO arrays at specific genomic loci and expressed fusion proteins (LacI-FKBP12, TetR-FRB, LacI-GFP, and TetR-mCherry) to induce and visualize chromosomal interactions upon rapamycin treatment, allowing accurate measurement of contact frequencies via time-lapse microscopy. They demonstrate that Hi-C can sensitively detect interactions occurring in as few as 7.8% of cells and exhibits a linear relationship between signal intensity and contact frequency across a wide range. Contrary to prior assumptions, Hi-C shows no inherent bias between intra- and inter-chromosomal interactions and static loops do not induce Topologically Associated Domains (TADs). These findings clarify the quantitative capabilities of Hi-C and suggest that previously reported biases likely stem from complexities in native chromatin dynamics rather than limitations of the assay itself.

The technical setup is thoughtfully designed, with clear rationale and implementation of the CICI system. The use of time-lapse microscopy to quantify absolute contact frequencies is a strong complement to the Hi-C data, and the mixing experiments to simulate varying contact frequencies are particularly elegant. The authors also provide sufficient methodological detail to enable reproducibility.

Overall, the manuscript is interesting and presents valuable information and resources for the field. The use of a synthetic, controllable system to benchmark Hi-C performance is innovative and provides a much-needed framework for interpreting Hi-C data quantitatively. The methodology is clearly described, and the results are well-supported by both imaging and sequencing data. This work will be of broad interest to researchers studying genome architecture and chromatin dynamics.

We thank the reviewer for the positive assessment of the paper.

Major comments:

1. Clarification on TAD Formation in Yeast

The authors should improve the interpretation of their data regarding loop formation and TADs. TADs, as classically defined in mammalian systems, are not typically observed in budding yeast, where chromatin is organized into much smaller self-interacting domains (PMID: 26119342). Given this, it is unclear why the authors expected that forcing a long-range loop in yeast would result in TAD formation. This expectation should be better justified or revised in the discussion.

We agree with the reviewer that canonical TADs in mammalian systems are generally not observed in budding yeast, likely due to the absence of CTCF and the inability to form stable cohesin-anchored loops. However, our study is motivated by a distinct question: *If a stable long-range loop is artificially imposed between two loci, can this lead to TAD formation?* Previous polymer simulations have suggested that the formation of a single static loop is not sufficient to generate TAD structures in Hi-C, but to our knowledge, this has not been directly tested *in vivo*. The CICI system allows us to induce a stable loop in a large fraction of cells, providing a unique experimental platform to test this theoretical prediction. Our result supports the polymer simulation that loop formation alone is NOT sufficient to drive TAD formation. We added this point to page 5.

2. Nature of the Synthetic Constructs

The TetO and LacO arrays used in this study are relatively large DNA elements (~3.6 kb and ~6.9 kb, respectively), which differ significantly from the short DNA motifs (e.g., CTCF binding sites) that mediate loop and TAD formation in mammalian genomes. These large synthetic arrays may not recapitulate the fine-scale architectural features of endogenous loop anchors. The authors should discuss how the size and nature of these constructs might influence the observed chromatin interactions and why the resulting contacts resemble long-range interactions seen in polycomb-mediated domains (PMID: 30008320; PMID: 31968256) or metaloops (PMID: 37536338).

The arrays used in our CICI strain is actually longer: 10.0 kb for LacO (256X) and 9.7 kb for TetO (192X), potentially allowing multivalent interactions among hundreds of proteins. CICI-based interactions is mainly generated through chromosome diffusion, which is fundamentally different from CTCF/ cohesin loops that rely on processive loop extrusion. Consistent with this, CICI does not produce TADs and readily forms both intra- and inter-chromosomal contacts, whereas CTCF-anchored loops are intra-chromosomal and lead to TAD formation.

The reviewer raised an interest point that our system may resemble some of the natural interactions mediated by polycomb domain and super enhancers. Like our arrays, these natural interacting domains are extended regions that can form long-range contacts through multi-valent interactions independently of CTCF/cohesin (Ogiyama *et al*, 2018; Rhodes *et al*, 2020). We added this point to page 5 in the results.

3. Extension to Mammalian Systems

While the yeast system is well-controlled and informative, the authors might consider extending this approach to mammalian or other metazoan cells where TADs are more clearly defined. Although this may be beyond the scope of the current study, such experiments would provide valuable insights into whether the synthetic interactions observed here behave more like polycomb-mediated contacts or canonical chromatin domains in higher eukaryotes.

We are indeed very interested in extending the CICI method into other systems. We are currently in the process of implementing CICI in mouse embryonic stem cells. This effort involves the integration

of two long DNA arrays and the expression of four fusion proteins, which must be carefully calibrated to achieve efficient CICI formation. Given the complexity of the system, this work is ongoing and will likely take at least 1-2 years to complete. As such, it falls beyond the scope of the current study, but we fully agree that applying CICI to metazoan systems will be insightful.

Minor Comments

4. Terminology Clarification: The manuscript occasionally uses the term "TAD" in the context of budding yeast, where such domains are not well-established. It would be helpful to clarify this distinction early in the text and consistently refer to "self-interacting domains" or "domain-like structures" when discussing yeast chromatin organization.

Related to point 1 above, we added the following sentence on page 5: "In mammalian cells, chromatin loops are often associated with TADs (8). TAD structures in budding yeast are less well defined (9), likely due to the absence of loop-anchoring factors such as CTCF. The strong static loops engineered by CICI could potentially promote the formation of "self-interacting domains" analogous to TADs".

5. Discussion of Limitations: While the authors briefly mention the advantages of the CICI system, a more explicit discussion of its limitations-such as the artificial nature of the constructs and potential off-target effects of rapamycin-would strengthen the manuscript.

Following the reviewer's suggestion, we added a more explicit discussion of the limitations of the CICI system in the revised Discussion section (page 7): "The main limitation of CICI is that it enforces loops via engineered interactions, and the resulting contact configuration and stability may differ from natural genomic interactions and bypass normal regulatory controls. In addition, CICI setup involves tuning the expression of four fusion proteins and integrating two repetitive arrays, which can be labor-intensive".

While the system relies on rapamycin-induced dimerization, we have not observed significant off-target effects of rapamycin in our rapamycin-resistant yeast strains. For example, there is no detectable changes in growth rate of the strains with or without rapamycin (Du *et al*, 2022). We therefore do not think this this a major concern of the method.

Bystricky K, Heun P, Gehlen L, Langowski J, Gasser SM (2004) Long-range compaction and flexibility of interphase chromatin in budding yeast analyzed by high-resolution imaging techniques.

Proc Natl Acad Sci U S A 101: 16495-16500

Du M, Zou F, Li Y, Yan Y, Bai L (2022) Chemically induced chromosomal interaction (CICI) method to study chromosome dynamics and its biological roles. *Nature communications* 13: 757

Imakaev M, Fudenberg G, McCord RP, Naumova N, Goloborodko A, Lajoie BR, Dekker J, Mirny LA (2012) Iterative correction of Hi-C data reveals hallmarks of chromosome organization. *Nat Methods* 9: 999-1003

Jusuf JM, Grosse-Holz S, Gabriele M, Mach P, Flyamer IM, Zechner C, Giorgetti L, Mirny LA, Hansen AS (2025) Genome-wide absolute quantification of chromatin looping. *bioRxiv*

Knight PA, Ruiz D (2012) A fast algorithm for matrix balancing. *IMA Journal of Numerical Analysis* 33: 1029-1047

Ogiyama Y, Schuettengruber B, Papadopoulos GL, Chang JM, Cavalli G (2018) Polycomb-Dependent Chromatin Looping Contributes to Gene Silencing during Drosophila Development. *Mol Cell* 71: 73-88 e75

Rhodes JDP, Feldmann A, Hernandez-Rodriguez B, Diaz N, Brown JM, Fursova NA, Blackledge NP, Prathapan P, Dobrinic P, Huseyin MK *et al* (2020) Cohesin Disrupts Polycomb-Dependent Chromosome Interactions in Embryonic Stem Cells. *Cell Rep* 30: 820-835 e810

Dear Dr. Bai,

Thank you for the submission of your revised manuscript. We have now received the enclosed report from referee 1 who was asked to assess it and I am happy to say that referee 1 supports the publication of your study now. S/he only has some more minor suggestions that I would like you to address and incorporate before we can proceed with the official acceptance of your manuscript.

A few editorial requests will also need to be addressed:

- The conflict of interest subheading needs to be corrected to "Disclosure and Competing Interests Statement" and needs to be placed after the Acknowledgments.
- The author credits need to be removed from the ms file. All credits need to be entered during online ms submission.
- The REFERENCE format needs to be corrected: it needs to be alphabetical, not numerical; et al needs to be used after 10 author names. Please use the EMBO reports reference style.
- The grant number T32 GM125592 is missing in our online submission system for NIH, please add. All funding information needs to be listed in the ms file and in our online submission system.
- A callout for Figure 2D is missing and "supplementary figure" is not a correct callout, please correct.
- The Materials and Methods section should include a separate Reagents and Tools Table file (listing key reagents, experimental models, software and relevant equipment and including their sources and relevant identifiers) and a Methods and Protocols section in which methods should be described using a step-by-step protocol format with bullet points. More information is available in our guide to authors online: <https://link.springer.com/journal/44319/submission-guidelines>
- Materials and Methods should be just Methods.
- There is a table on page 7 that needs to be labeled and called out in the ms text.
- Our routine image analysis detected a possible data re-use without reference in the figure legend between Figure 2A and Figure 3B, as well as between Figure 2A and Figure 4B. Can you please clarify what happened?
- Please note that the specific URL for the GSE283767 dataset needs to be provided in the data availability section.

EMBO press papers are accompanied online by A) a short (1-2 sentences) summary of the findings and their significance, B) 2-3 bullet points highlighting key results and C) a synopsis image that is exactly 550 pixels wide and 200-600 pixels high (the height is variable). The synopsis image should provide a sketch of the major findings, like a graphical abstract. Please note that text needs to be readable at the final size. Please send us this information along with the final manuscript.

Referee #1:

Overall the authors have done a good job addressing the comments of all reviewers.

One issue needs to still be clarified: the authors state that they only count stable associations, excluding random collision. However, Hi-C will detect both types of interaction. Can the authors test whether including both types changes the results, or perhaps even alters the difference in slopes that they now observe?

The revised manuscripts include a large sample size, demonstrating the reproducibility and generalizability of the conclusions. I

have no further questions. I only have one minor suggestion:

The authors may consider overlaying the $P(s)$ slope from yeast Hi-C data on top of panel 3D, enabling a direct comparison that the decay of contact probability with genomic distance between the reconstructed HI-C contacts and the real signal obtained from bulk Hi-C assays.

Further comments from referee 1:

I think the authors address my comment well. It may be advisable when they explicitly state in the paper the caveat that they focus only on stable interactions seen in the microscope, while Hi-C sees all interactions. With that, the manuscript can be accepted, in my view.

Here is a summary of the changes made to the manuscript in response to editorial requests and the to the referee #1's comment.

1. We reformatted the manuscript in terms of "Disclosure and Competing Interests Statement", author credit, references, grant information, figure citation, and data URL.
2. We added a separate Reagents and Tools Table in the manuscript.
3. The plasmid and strain list is now labeled as Table EV1 and cited in Methods.
4. Our routine image analysis detected a possible data re-use without reference in the figure legend between Figure 2A and Figure 3B, as well as between Figure 2A and Figure 4B. Can you please clarify what happened?

Thank you for flagging this. We do not have a Figure 4B in the manuscript, so we suspect the reference is to Figure EV1. To clarify, Figure 2A shows the Hi-C maps for the 0% (-rapamycin baseline) and 100% (+rapamycin) conditions. Figure 3B and Figure EV1 present Hi-C data for mixed populations generated by combining these two conditions at defined ratios. The 0% and 100% endpoints are shown in Figure 3B/EV1 again as reference points. We have now explicitly stated this in the figure legends for Figure 3B and Figure EV1, noting that the 0% and 100% panels are reused from Figure 2A.

5. A short (1-2 sentences) summary of the findings and their significance: Hi-C reports relative, rather than absolute, contact frequencies. Using chemically induced chromosomal interactions (CICI), this study calibrates Hi-C signal strength against absolute contact frequency and provides a framework for quantitative interpretation of Hi-C data.
6. 2-3 bullet points highlighting key results:
 - Strong, static intra-chromosomal loops increase local proximity only over tens of kilobases without generating topologically-associated domains.
 - Hi-C signal scales linearly with absolute contact frequencies.
 - Hi-C shows no intrinsic preference for intra- over inter-chromosomal contacts.
7. A synopsis image is included in this submission.
8. To address the reviewer's comments, we (1) added Figure EV2, which shows the Hi-C signal after background subtraction to remove contacts arising from random polymer collisions, and (2) updated Figure 3D to plot absolute contact frequency as a function of genomic distance.

Lu Bai
Penn State University
Department of Biochemistry and Molecular Biology, Department of Physics
406 South Frear
University Park, PA 16802
United States

Dear Dr. Bai,

I am very pleased to accept your manuscript for publication in the next available issue of EMBO reports. Thank you for your contribution to our journal.

You may qualify for financial assistance for your publication charges - either via a Springer Nature fully open access agreement or an EMBO initiative. Check your eligibility: <https://link.springer.com/journal/44319/how-to-publish-with-us>

>>> Please note that it is EMBO Reports policy for the transcript of the editorial process (containing referee reports and your response letter) to be published as an online supplement to each paper. If you do NOT want this, you will need to inform the Editorial Office via email immediately. More information is available here: <https://link.springer.com/partners/embo-press/editorial-policies#Peer%20review>